

# Technical note: Literature based approach to estimate future snow

Bettina Richter[1] and Christoph Marty[1]

[1]WSL Institute for Snow and Avalanche Research SLF, Davos, Switzerland

**Correspondence:** Bettina Richter (bettina.richter@slf.ch)

**Abstract.**

The seasonal snow cover in the European Alps is increasingly threatened by rising temperatures due to climate change. Still, downscaled climate projections are lacking for many regions. To address this gap, we developed a literature-based approach for projecting future snow depths, that is applicable to all locations where historical snow depth data is available.

We harmonized heterogeneous literature on future snow depth and snow water equivalent by translating emission scenarios to corresponding temperature scenarios and standardizing seasonal periods. Then, we parameterized localized reduction curves based on elevation, temperature scenarios and local climatologies, as mean snow cover length and mean maximum snow depth. This method was applied to four measurement stations in Switzerland under a $+2°$ C temperature scenario, revealing significant declines in snow depth and season length, especially at lower elevations. Validation against published data shows that the approach captures key trends in snow loss, despite the simplification of climate dynamics.

This resource-efficient method provides a practical tool for estimating climate change related snow depth declines in snow dominated regions, which are lacking highly resolved climate projections, and can support decision-makers in developing adaptation strategies for climate-related challenges.

## 1 Introduction

Seasonal snow cover plays a crucial role in Alpine hydrology, ecology, and winter tourism. In the context of ongoing climate change, it is increasingly threatened by rising temperatures. Both, snow depth and snow water equivalent (SWE) have shown substantial decreases across the European Alps in recent decades (Marty et al., 2025; Ranzi et al., 2024; Broust et al., 2024), with future projections indicating further declines across all ranges of elevations and regions (Kotlarski et al., 2022; Bülow et al., 2025). While detailed climate projections exist for selected study areas and provide detailed insights into future snow cover (Marty et al., 2017; Schmucki et al., 2015; Willibald et al., 2020; Verfaillie et al., 2018), many Alpine regions still lack high resolution projections. This presents a challenge for practitioners and decision-makers requiring localized snow cover information to support climate adaptation strategies.

To bridge this gap within a short-term project with limited resources, we developed a transferable, literature-based approach, which is applicable to climatological datasets (both point-based and gridded) by synthesizing existing studies to estimate future snow cover changes.



## 2   Methods

### 2.1   Synthesizing literature values

The heterogeneity of available studies posed several methodological challenges, e.g. the heterogeneity in projected regions and elevations, emission scenarios and examined variables.

Reported variables ranged from decreases in seasonal means for different period (Willibald et al., 2020; Kotlarski et al., 2022; Morin et al., 2018; Marty et al., 2017; Verfaillie et al., 2018), season lengths for different thresholds in snow depths (Willibald et al., 2021; Verfaillie et al., 2018; Morin et al., 2018), monthly values (Bülow et al., 2025; Marty et al., 2017) or seasonal evolutions (Schmucki et al., 2017; Schmucki, 2015; Fiddes et al., 2022) of either SWE or snow depths. Since the relative reductions in SWE and snow depth were comparable across these studies (Schmucki et al., 2015; Verfaillie et al.,

2018), we treated the reduction values as interchangeable.

Tables A1 and A2 in the appendix summarize the examined variables. These studies were generally divided into two groups: Literature-Fit and Literature-Validation (see comments in Tables A1 and A2). The data in Literature-Fit provided daily or monthly snow depths, both for the reference period and the future projection, which could be used to train the reduction curves in Section 2.2. The data in Literature-Validation reported seasonal reduction values or decreases in season length and was used

for validation only (Sections 2.4 and 3.3).

#### 2.1.1   Translating emission scenarios to temperature scenarios

Most studies referred to different emission scenarios (e.g., various RCPs). To address this complexity, we standardized reported climate change scenarios by translating all RCPs, reference periods, and projected periods into corresponding temperature scenarios, if temperature scenario was not directly reported. To this end, we used the reports CH2011 and CH2018 (CH2018, 2018; CH2011, 2011) and summarized emission and corresponding temperature scenarios in Table B1 (see Appendix).

#### 2.1.2   Synthesizing different seasonal means

Reduction values for seasonal means in the Literature-Validation dataset were reported for various time periods, ranging from the shortest—December to February (DJF, 3 months)—to the longest—September to August (SONDJFMAMJJA, 12 months). For consistency, we synthesized all results to a common six-month period from November to April (NDJFMA).

To this end, we used the Literature-Fit dataset, which was linearly interpolated to daily values, to calculate mean snow depths for both the reference period ($HS_{ref}$) and the future projection ($HS_{fut}$) across different seasonal windows (e.g., NDJFMA). The relative decrease in mean snow depth for NDJFMA was then calculated as:

$$\text{NDJFMA-decrease} = \frac{HS_{fut} - HS_{ref}}{HS_{ref}} \tag{1}$$

Reduction values for other periods were computed analogously and compared to the NDJFMA-decrease (Figure 1). Depend-

ing on the specific period and elevation, deviations of up to 10% from the NDJFMA-decrease were observed. To account for




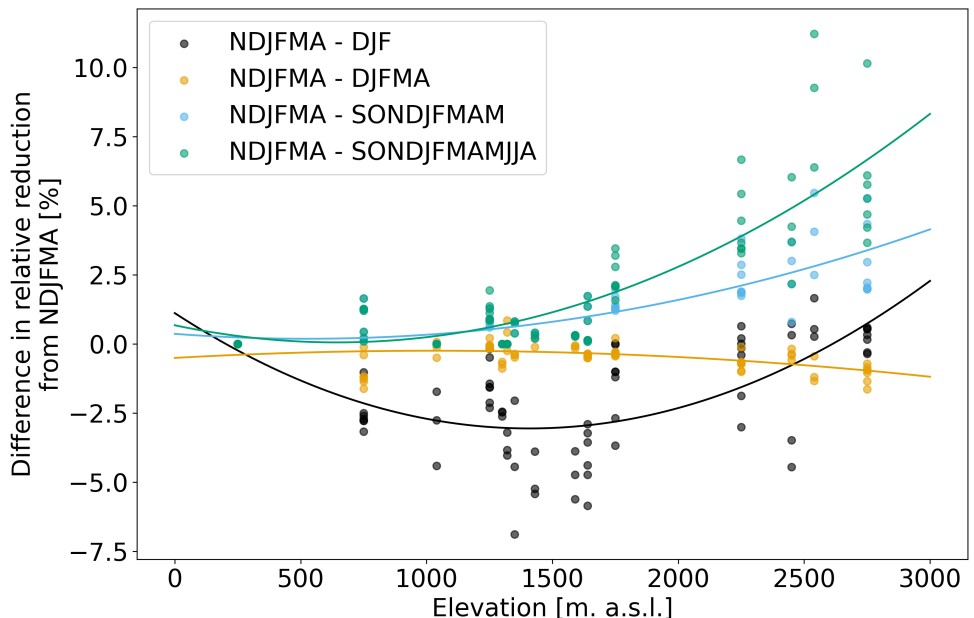

**Figure 1.** Relationship of relative decreases in mean snow depths for different periods (colors) compared to NDJFMA period. Values were derived from studies which provided monthly or daily snow depths and decreases. For each period we computed a second order polynomial fit with elevation.

these variations, we applied a second-order polynomial fit to adjust reduction values based on both period length and elevation (lines in Figure 1).

In general, relative decreases in mean snow depth were more pronounced for longer periods. For example, at 2500 m elevation, if a SONDJFMAMJJA-decrease of -25% was reported, this would compare to a NDJFMA-decrease of -20% on average
(green line in Figure 1).

### 2.2    Fitting of Seasonal Reduction Curves

We analyzed the seasonal snow depth reduction curves for the Literature-Fit data, by first linearly interpolating reported values to daily data. If reduction values were not provided, we computed those from reference and future snow depth or SWE data.

Figure 2 (top) shows reference and future snow depth at 1350 m a.s.l. under a +2.4 °C scenario from Schmucki (2015).
Peak snow depth decreased from 116 cm to 52 cm, with a reported reduction in season length of approximately 20 days. The corresponding future relative snow depth, which are equivalent to relative reduction curves, (Fig. 2, bottom) peaked shortly before the seasonal maximum, and approached 0 % toward both ends of the season. This behavior implies a delay in season start and an earlier season end, reflecting a general shortening of the snow season. All relative reduction curves consistently





showed this behavior without any systematic asymmetry between accumulation and ablation periods. Therefore, reduction
curves $f_{red}(x)$ could be well approximated by a quadratic function:

$$f_{red}(x) = -100 + a - \frac{a}{c^2} \cdot (x - b)^2, \quad \text{with } f(x) \geq -100, \tag{2}$$

To express the relative future snow depth $f(x) = f_{red}(x) + 100$ directly, this can be rewritten as:

$$f(x) = a - \frac{a}{c^2} \cdot (x - b)^2, \quad \text{with } f(x) \geq 0, \tag{3}$$

where:

– $x$ is the day of water year (DOWY, ranging from 1 to 366, with DOWY 1 = 1. September),

– $a$ denotes the maximum of $f(x)$, corresponding to the highest relative snow depth in future,

– $b$ is the day of the year (DOWY) on which this maximum occurs,

– $c$ defines half the width of the curve, and thus approximates half the snow season length (i.e., days with snow on the
  ground before and after $b$).

As season lengths can highly differ between regions and locations, instead of finding trends for $b$ and $c$, we compared the
days between peak reduction and peak snow depth $\Delta b$ and the fraction of future season length compared to reference season
length $\Delta c$:

$$\Delta b = b - \text{DOWY}(HS_{ref,max}) \text{ [days]} \tag{4}$$

$$\Delta c = \frac{len(HS_{fut} > 0\text{cm})}{len(HS_{ref} > 0\text{cm})} \cdot 100 \text{ [\%]} \tag{5}$$

In Figure 2, the maximum future snow depth $a = 47\%$ and the reduction curve peaks on $b = 144$ DOWY (corresponding to
23. January), hence the peak of the reduction curve is 41 days prior to the peak in reference snow depth ($\Delta b = -41$). The total
snow duration decreases from 216 days to 195 days, resulting in a relative decrease of $\Delta c = 90\%$.

Finally, we compared all reduction parameter ($a$, $\Delta b$, and $\Delta c$) from the Literature-Fit data with temperature scenario and
elevation (see Section 3.1) and trained a linear regression model. To avoid overfitting, we used linear terms of temperature
scenario and elevation and their interaction terms (elevation $\times \Delta T$), which were scaled before training. While the reduction
parameters $a$, $\Delta b$ and $\Delta c$ can be computed for any given elevation and temperature scenario, these parameters were trained on
data with elevations ranging from 750 m and 2750 m and temperature scenarios from $+1.1°C$ to $+4.8°C$ and should be treated
with caution outside these ranges.

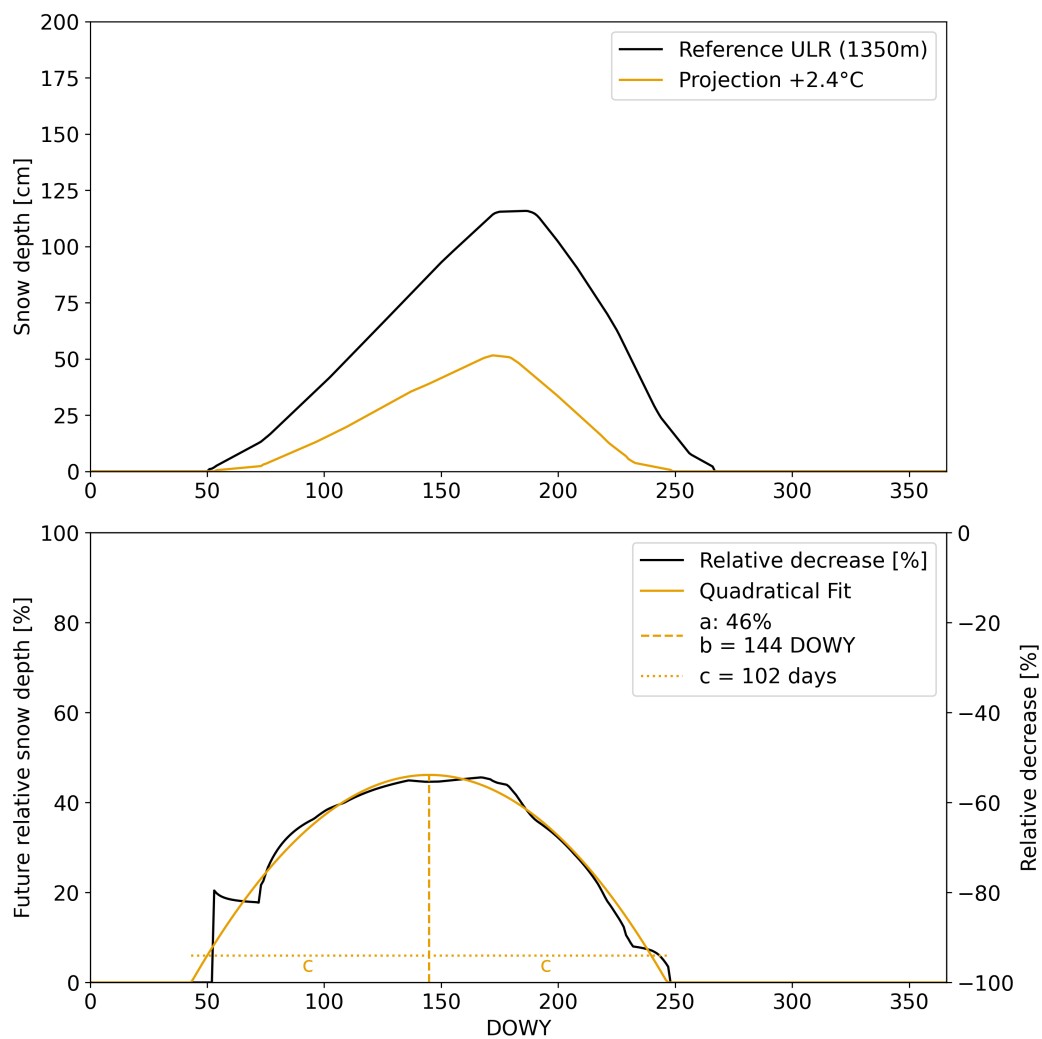

**Figure 2.** (Upper) Reference and future snow depth from Schmucki (2015) (temperature scenario: $+2.4\,°\mathrm{C}$, elevation: $1350\,\mathrm{m}$ a.s.l., monthly averages) and (lower) corresponding future relative snow depths. Orange line shows the quadratic fit for the reduction curve. x-axis is the day of water year (DOWY) starting on 1. September.



## 2.3 Apply reduction curve to project future snow evolution

We applied the reduction curves to retrieve future snow depths for four measurement stations in Switzerland: Weissfluhjoch (WJ, 2540 m a.s.l.), in the eastern Swiss Alps, Maloja (MA, 1810 m a.s.l.), in the southern Swiss Alps, Saanenmöser (SM, 1390 m a.s.l.), in the western Swiss Alps, and Engelberg (EN, 1023 m a.s.l.), in the central Swiss Alps. These stations provide daily manually measured snow depth data from winter season 1991-2020 (30 years). We computed daily median, as well as the 5th and the 95th percentile for those stations, as our reduction curves were trained on data, which does not account for extreme events.

We chose a temperature scenario of $\Delta T = +2°C$ for the projections then computed the reduction parameters $a$, $\Delta b$ and $\Delta c$ for the given elevations. As our reference period 1991-2020 experienced a mean annual temperature increase of $+0.5°C$ compared to the period 1981-2010 (Senoner et al., 2023), this temperature scenario of $\Delta T = +2°C$ refers to the climate period 2043-2072 for the RCP8.5 scenario of CH2018 (2018).

Climate projections show uncertainties in temperature scenarios of around $+/-1°C$ for each RCP-scenario between years 2000 and 2100 (CH2018, 2018). Therefore, we applied the same uncertainty range for the snow projections as follows:

- Temperature scenario $\Delta T$ for median snow depths (here: $\Delta T = +2°C$)

- "$\Delta T + 1°C$" scenario used for 5th percentile (higher increase, more pessimistic scenario, here: $+3°C$)

- "$\Delta T - 1°C$" scenario used for 95th percentile (less increase, more optimistic scenario, here: $+1°C$)

To apply the reduction curve (Equation 3) to climatological evolutions we have to compute $b$ and $c$ from $\Delta b$ and $\Delta c$, respectively. To this end, we first determined $\text{DOWY}(HS_{ref,max})$ and the reference season length $len(HS_{ref} > 0\text{cm})$. We suggest to smooth reference snow evolutions prior to determining $\text{DOWY}(HS_{ref,max})$ using a running mean of 30 days, to smoothen temporal variability in the data. Then $b$ and $c$ can be computed as followed:

$$b = \text{DOWY}(HS_{ref,max}) + \Delta b \tag{6}$$

$$c = \frac{len(HS_{ref} > 0\text{cm}) \times \Delta c}{2} = \frac{len(HS_{fut} > 0\text{cm})}{2} \tag{7}$$

Finally, using the reduction curve (Equation 3) and reference snow depths $HS(x)_{ref}$, future snow depths $HS(x)_{fut}$ can be computed as follows:

$$HS(x)_{fut} = f(x) \cdot HS(x)_{ref} \tag{8}$$

## 2.4 Variables used for validation

After synthesizing all seasonal decreases from the Literature-Fit and Literature-Validation dataset into NDJFMA-decreases (Section 2.1.2), we compared these Literature values to the NDJFMA-decreases from our projections of the four stations using Equation 1.





Furthermore, we also looked at the relative reduction in season lengths by counting the days for which a certain snow depth was reached, both in the reference $len(HS_{ref} > x)$ and for the projections $len(HS_{fut} > x)$. The relative reduction in season length $(HS > x)$-decrease was then computed as follows:

$$(HS > x)\text{-decrease} = \frac{len(HS_{fut} > x) - len(HS_{ref} > x)}{len(HS_{ref} > x)} \cdot 100 \qquad (9)$$

The Literature-Validation dataset contains decreases in season lengths for the following thresholds: $> 5\,\text{cm}$, $> 30\,\text{cm}$, $>$

$50\,\text{cm}$ and $> 100\,\text{cm}$. As most values were reported for the former two thresholds, we chose the following variables for validation:

- Relative decrease in mean November-April snow depth (NDJFMA-decrease).

- Relative decrease in season length with more than $30\,\text{cm}$ snow on the ground $(HS > 30\text{cm})$-decrease.

- Relative decrease in season length with more than $5\,\text{cm}$ snow on the ground $(HS > 5\text{cm})$-decrease.

Projected relative decreases were calculated for the median snow depths and 5-95 percentiles.

## 3    Results

### 3.1    Reduction curve parameters with respect to temperature and elevation

Figure 3 (left) presents the reduction parameter $a$, which corresponds to the maximum future relative snow depth. As expected, $a$ decreases with increasing temperature scenarios, indicating less snow under warmer conditions. Furthermore, $a$ increases

with elevation, indicating that the decrease of future snow depth is more pronounced at lower elevations.

Although no consistent trend was observed for the parameter $b$ itself, $\Delta b$ were predominantly negative (Figure 3, middle), suggesting a shift in the timing of peak snow depth towards earlier in the season. Notably, this shift becomes more pronounced with elevation, as $\Delta b$ decreases at higher altitudes.

The relative change in season length $\Delta c$ is shown in Figure 3 (right). All studies indicated shorter snow seasons in fu-

ture scenarios, with the reduction in season length being more substantial at lower elevations and under higher temperature scenarios.

The following regression formulas were derived to describe the relationships of the reduction parameters with temperature and elevation (lines in Figure 3):

$$a = 83.51239 - 23.89164 \cdot \Delta T + 0.01085 \cdot h + 0.00463 \cdot \Delta T \cdot h, \quad \text{with } a \in [0, 100] \qquad (10)$$


$$\Delta b = 16.28888 - 1.61312 \cdot \Delta T - 0.02390 \cdot h - 0.00094 \cdot \Delta T \cdot h \qquad (11)$$




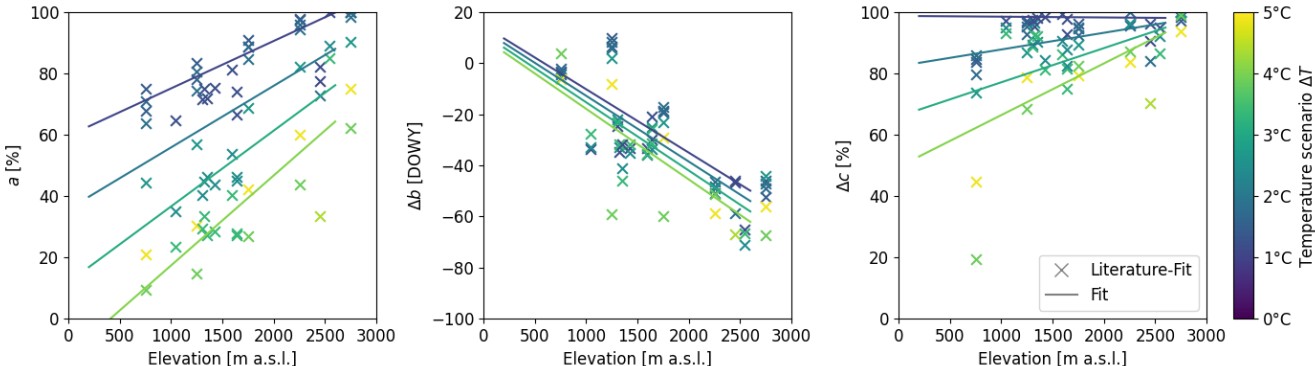

**Figure 3.** Reduction parameters from the Literature-Fit dataset and corresponding fits with elevation and temperature scenario, which were used for computing reduction curves for future snow projections: (left) maximum of the future relative snow depth $a$, (middle) days between peak reduction and peak snow depth $\Delta b$, and (right) the future relative season length $\Delta c$.

$$\Delta c = 115.18564 - 16.41657 \cdot \Delta T - 0.00595 \cdot h + 0.00570 \cdot \Delta T \cdot h \tag{12}$$

### 3.2 Projected snow evolutions

Snow projections under a $\Delta T = +2°C$ temperature scenario indicate a decline in snow depth across all sites (Figure 4). At Weissfluhjoch, the peak median snow depth decreases from 215 cm to 171 cm, while in Saanenmöser it drops from 64 cm to 36 cm. In Engelberg, the median snow depth during the reference period never exceeded 30 cm, and in 5% of winters, snow depth remained at 0 cm throughout the entire year.

All projections also indicate shorter snow seasons in the future. For example, at Weissfluhjoch, the snow season (with 160    $HS > 0$ cm) is projected to begin approximately two weeks later and to end nearly two weeks earlier on average.

### 3.3 Validation of snow projections

We validated the projections by comparing the projected decreases at all four study sites with values reported in the literature (Figure 5). Projected decreases shown as bars represent the 5th and 95th percentiles, while black lines indicate the decrease in median snow depth. Since literature values were not always based on the exact same temperature scenarios or elevations, 165    comparisons were quantitatively made based on the range of values rather than exact matches.

Both the Literature-Fit and Literature-Validation datasets show similar trends (percentage changes) for the NDJFMA-decrease, the (HS> 30 cm)-decrease and the (HS> 5 cm)-decrease: the relative decreases are stronger with higher temperature scenarios and at lower elevations (Figure 5). The projections align well with the reported ranges and replicate the expected elevation-dependent trends. Both, literature data and projections show weaker (HS> 5 cm)-decreases compared to the 170    (HS> 30 cm)-decrease.



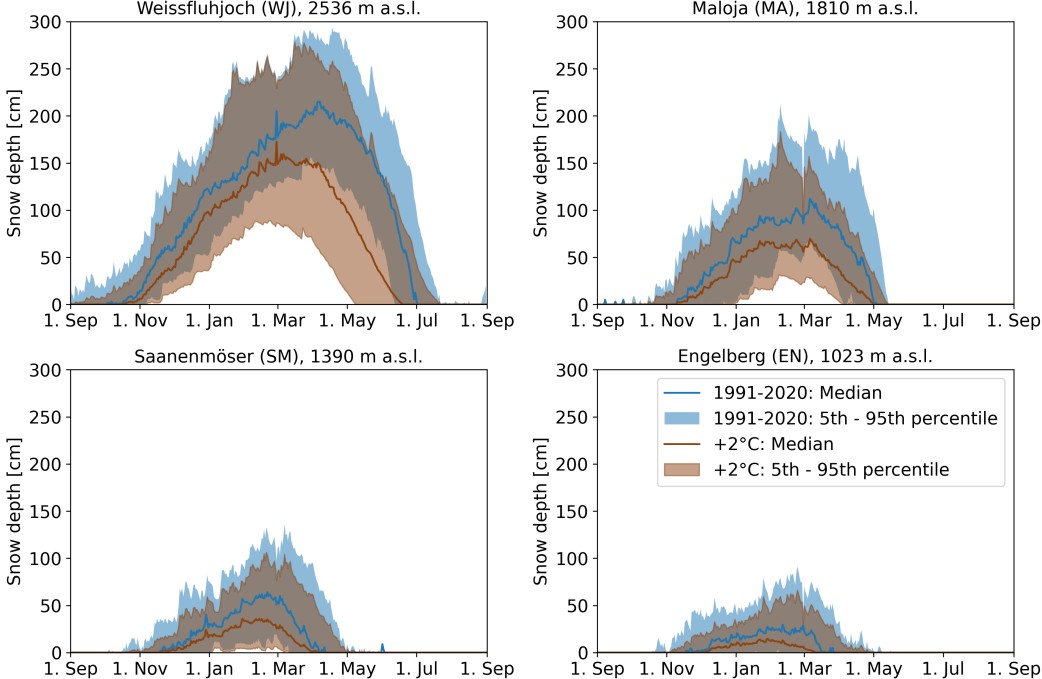

**Figure 4.** Snow depth evolution for Weissfluhjoch (upper left), Saanenmöser (upper right), Maloja (lower left) and Engelberg (lower right) for the reference period 1991-2020 and a future projection for the temperature scenario $+2°C$.

Uncertainty ranges in the projections, showing the relative decrease in the 5th and the 95th percentile, seemed to be occasionally larger than uncertainty ranges from literature values. This is due to the methodology (see Section 2.3), as we are projecting median and 5-95 percentiles rather then individual years: For instance, the 5th percentile snow depth at Engelberg (EN) is zero at each calendar day in the reference period (see Figure 4). This indicates that there is currently not one period (day) throughout a year, where snow on the ground can be guaranteed in Engelberg. As such the projected decrease in Figure 5 was set to -100%. On the other hand, we want to highlight that this does not imply that 5% of the future winters will be entirely snow-free.

## 4  Discussion

This study presents a practical and efficient alternative to high-resolution climate model downscaling for estimating future changes in snow depth and season length. Synthesizing existing literature and applying a simplified yet structured method to observational snow data allows to assess climate impacts in regions where localized projections are scarce or unavailable.

A major strength of this method lies in its low resource demand. No computational complex physically-based snow models or dynamically downscaled regional climate simulations are required. Instead, observed snow data in combination with




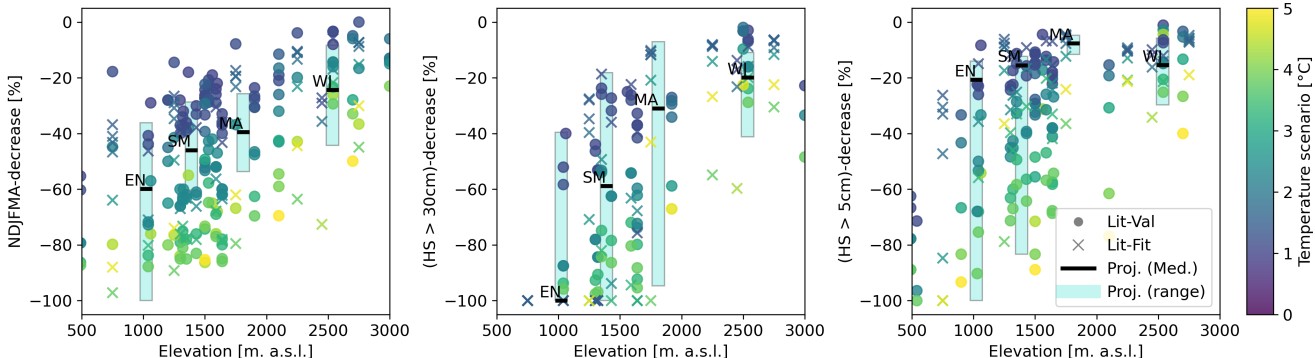

**Figure 5.** Projected (bars) and literature based (markers) values for (left) the relative decrease in mean November to April snow depths NDJFMA-decrease, (middle) relative decrease in season length with more than 30 cm snow (HS> 30 cm)-decrease and (right) decrease in season length with more than 5 cm snow (HS> 5 cm)-decrease with elevation. Colors indicate temperature scenarios. Projections show median decrease (black line) and 5th and 95th percentile (bars) for the four stations Weissfluhjoch (WJ), Maloja (MA), Saanenmöser (SM), and Engelberg (EN).

parameterized reduction curves is sufficient to obtain snow projections for different climate scenarios. As such, the approach
inherently captures local climatology and site-specific features (e.g., exposure, shading, cold pool valleys) that are often missed
in coarse-grid climate models (Frei, 2014). This is especially valuable for metrics like snow season length above a specific
threshold (e.g., $HS > 30$ cm), which are highly sensitive to local topography and micro-climatic conditions.

Despite these strengths, the approach has several important limitations, in the first place the simplification of climate dy-
namics. While temperature change is the dominant driver of future snow loss in the Alps (Marty et al., 2017; Verfaillie et al.,
2018; Kotlarski et al., 2022), other climatological uncertainties such as changing precipitation, or change of large scale weather
patterns are not explicitly incorporated. These factors are only indirectly captured, as the reduction parameters were trained on
projections that do consider such dynamics. Still, this indirect inclusion does not allow assessing their future contributions.

Furthermore, the quadratic approximation of snow reduction curves and the linear interpolation of reduction parameters with
temperature and elevation may not capture the full complexity of snow accumulation and melt processes. These simplifications
may limit accuracy in regions with strong seasonal asymmetries or where snow dynamics are highly variable.

The reduction parameters where trained for elevations ranging between 750 m – 2750 m and temperature scenarios between
$+1.1°$ C and $+4.8°$C. Extrapolations outside these ranges should be treated with caution.

The literature-derived values often lack clear definitions of ensemble spreads, e.g. spread of mean of individual ensemble
members (Schmucki et al., 2015) or spread of all modeled snow depth evolutions. As a robust statistical assessment was
not possible, we adopted a simple $+/-1°$C variation around the central temperature scenario to approximate uncertainty
bounds (5th and 95th percentiles). Although this is a coarse approximation, the resulting projection spread remains within the
variability reported across different studies (Marty et al., 2017; Bülow et al., 2025).



However, while the approach offers high transferability and ease of use, it simplifies complex climatic and geographic inter-actions. Thus, it is best used as a first-order estimate in regions where detailed projections are not available, or to complement more detailed modeling efforts.

## 5 Conclusion

This study introduces a resource-efficient approach to project future snow cover evolution across Alpine regions. Literature values were used to obtain localized reduction curves depending on temperature scenario and elevation. The relative reductions curves could be described with three metrics: (1) maximum relative future snow depth, (2) shift in the timing of peak snow depth, and (3) relative shortening of the snow season. Projections show consistent and plausible snow depth trends for different elevations and temperature scenarios.

This method is rather simplified and rudimentary, still it offers a robust and adaptable framework for estimating future local or regional snow depth changes in the absence of high-resolution climate projections. As such, it provides a valuable tool for the assessment of climate impacts and development of adaptive strategies in snow-dependent regions of the European Alps.

*Code and data availability.* Code and literature data will be uploaded to envidat.ch (URL will be provided). In-situ snow depth data from SLF stations can be freely downloaded from: https://www.slf.ch/en/services-and-products/slf-data-service.

.

**Appendix A: Table of Literature values used for training and validation**





| Literature | Reports Reference period | Scenario | Climate period | Elevation a.s.l. [m] | Region | Comments |
|---|---|---|---|---|---|---|
| Bülow et al. (2025)* | CH2018 1971-2000 | RCP26 RCP45 RCP85 | 2021-2050 2069-2098 | 0-500 500-1000 1000-1500 1500-2000 2000-2500 2500-3000 | Alps | Literature-Fit monthly values |
| Marty et al. (2017) | CH2011 1980–2009 | A2 | 2071-2100 | 0-500 500-1000 1000-1500 1500-2000 2000-2500 2500-3000 | Region Aare | Literature-Fit monthly values |
| Schmucki et al. (2017) Schmucki (2015) | CH2011 1984–2010 | A1B | 2020-2049 2045-2074 2070-2099 | 2540 1640 1640 1590 1430 1350 1320 1300 1040 | Weissfluhjoch San Bernardino Zermatt Davos Montana Ulrichen Adelboden Scuol Engelberg | Literature-Fit daily values |
| Fiddes et al. (2022) | CH2018 1981–2010 | RCP2.6 RCP8.5 | 2031-2060 2070-2099 | 2450 m | IMIS | Literature-Fit daily values |
| Willibald et al. (2020) | CH2018 1980-2009 | RCP8.5 | 2010-2039 2040-2069 2070-2099 | 2540 1640 1590 1430 1350 1320 1300 1040 | Weissfluhjoch Zermatt Davos Montana Ulrichen Adelboden Scuol Engelberg | Literature-Validation Nov-Apr |
| Willibald et al. (2021) | CH2018 1980-2009 | RCP8.5 | 2010-2039 2040-2069 2070-2099 | 2540 1590 1040 | Weissfluhjoch Davos Engelberg | Literature-Validation season length (HS> 30cm) |
| Verfaillie et al. (2018) | CH2018 1986–2005 | RCP2.6 RCP4.5 RCP8.5 | 2022-2038 2042-2058 2062-2078 2082-2098 | 1500 | Chatreuse | Literature-Validation Dec-Apr season length (HS> 50 cm) |

**Table A1.** Overview over Literature values, which were used for training (Literature-Fit, first four studies in Table A1) and validation (Literature-Validation). Studies which reported values for SWE were marked with *. (Part 1, part 2 see Table A2)



| Literature | Reports Reference period | Scenario | Climate period | Elevation a.s.l. [m] | Region | Comments |
|---|---|---|---|---|---|---|
| Schmucki et al. (2015) | CH2011 1984–2010 | A1B A2 | 2020-2049 2045-2074 2070-2099 | 2540 1640 1640 1590 1430 1350 1320 1300 1040 | Weissfluhjoch San Bernardino Zermatt Davos Montana Ulrichen Adelboden Scuol Engelberg | Literature-Validation Dec-Feb season length (HS> 30 cm) |
| Kotlarski et al. (2022)* | CH2018 1981-2010 | RCP2.6 RCP4.5 RCP8.5 | 2070-2099 | 0-500 500-1000 1000-1500 1500-2000 2000-2500 2500-3000 | Alps | Literature-Validation Sep-May |
| Kotlarski et al. (2022) | CH2018 1981-2010 | RCP2.6 RCP4.5 RCP8.5 | 2021–2094 | 1200 2100 3000 | Mont-Blanc | Literature-Validation Nov-Apr |
| Kotlarski et al. (2022) | CH2018 1971-2000 | RCP2.6 RCP4.5 RCP8.5 | 2021–2050 2070–2099 | 1920 2500 3000 | Ötztaler Alps | Literature-Validation season length (HS> 30cm) |
| Marty et al. (2017) | CH2011 1999-2012 | A1B A2 RCP3PD | 2020–2049 2045–2074 2070–2099 | 1530 1903 | Aare Graubünden | Literature-Validation Sep-Aug |
| Marty et al. (2017) | CH2011 1999-2012 | A2 | 2020–2049 2045–2074 2070–2099 | 540 1030 1650 | Bern Grindelwald Mürren | Literature-Validation season length (HS> 5cm) |
| Marty et al. (2017) | CH2011 1999-2012 | A2 | 2020–2049 2045–2074 2070–2099 | 3000 | Aare | Literature-Validation season length (HS> 30cm) |
| Morin et al. (2018) | IPCC2013 1986-2005 | RCP2.6 RCP4.5 RCP8.5 | +1.5°C +2°C +3°C +4°C +5°C | 1500 2100 2700 | Mont-Blanc | Literature-Validation Dez-Apr |
| Morin et al. (2018) | IPCC2013 1986-2005 | RCP2.6 RCP4.5 RCP8.5 | +1.5°C +2°C +3°C +4°C +5°C | 900 1500 2100 2700 | Pyrenees | Literature-Validation season length (HS> 5cm) |

**Table A2.** Overview over Literature values, which were used for training (Literature-Fit, first four studies in Table A1) and validation (Literature-Validation). Studies which reported values for SWE were marked with *. (Part 2, part 1 see Table A1)



## Appendix B: Table: Linking RCP-Scenarios and Climate periods to temperature scenarios

*Author contributions.* BR: Data analysis, Conceptualization, Methodology, Software, Writing – original draft. CM: Conceptualization, Methodology, Writing – review & editing.

*Competing interests.* The authors declare that they have no conflict of interest.

*Acknowledgements.* The authors want to thank the State Secretariat for Economic Affairs SECO and 'Seilbahnen Schweiz' for funding the project. Additional financial support was provided by Speed2Zero. We also thank Sven Kotlarski and Katharina Bülow for their valuable
contributions in providing data and insightful discussions.





**Table B1.** Mapping of emission scenario, reference and climate period to temperature scenario (ΔT).

| Report | Emission scenario | Reference period | climate period | ΔT [°C] |
|--------|-------------------|------------------|----------------|---------|
| CH2011 | A1B | 1984–2010 1999–2012 | 2020–2049 | +1.2 |
| | | | 2045–2074 | +2.4 |
| | | | 2070–2099 | +3.3 |
| | A2 | 1984–2010 1999–2012 1980–2009 | 2020–2049 | +1.1 |
| | | | 2045–2074 | +2.3 |
| | | | 2070–2099 | +3.8 |
| | | | 2071–2100 | +3.8 |
| | RCP3PD | 1999–2012 | 2020–2049 | +1.2 |
| | | | 2045–2074 | +1.2 |
| | | | 2070–2099 | +2.4 |
| CH2018 | RCP2.6 | 1971–2000 | 2021–2050 | +1.5 |
| | | | 2069–2098 | +1.6 |
| | | 1981–2010 | 2031–2060 | +1.1 |
| | | | 2070–2099 | +1.2 |
| | | 1986–2005 | 2024–2037 | +0.5 |
| | | | 2044–2057 | +0.8 |
| | | | 2064–2077 | +0.9 |
| | | | 2084–2097 | +0.8 |
| | RCP4.5 | 1971–2000 | 2021–2050 | +1.7 |
| | | | 2069–2098 | +2.7 |
| | | 1986–2005 | 2024–2037 | +0.8 |
| | | | 2044–2057 | +1.4 |
| | | | 2064–2077 | +1.8 |
| | | | 2084–2097 | +1.9 |
| | RCP8.5 | 1971–2000 | 2021–2050 | +1.8 |
| | | | 2069–2098 | +4.8 |
| | | 1981–2010 | 2031–2060 | +1.9 |
| | | | 2070–2099 | +4.4 |
| | | 1980–2009 | 2010–2039 | +1.0 |
| | | | 2040–2069 | +2.4 |
| | | | 2070–2099 | +4.3 |
| | | 1986–2005 | 2024–2037 | +0.8 |
| | | | 2044–2057 | +1.7 |
| | | | 2064–2077 | +3.0 |
| | | | 2084–2097 | +4.1 |



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
