# Peer review of "Technical note: Literature based approach to estimate future snow"

_EGUsphere, 2025_

## Author Comment (AC1)

**Author responses to reviewer 2 comments on:**

**"Technical note: Literature based approach to estimate future snow"**

by Richter et al. in *Hydrology and Earth System Sciences* (*HESS*)

We thank the reviewer for the time to assess our work and for the valuable feedback and suggestions. We respond to each point of the reviews below. The reviewer comments are highlighted in blue while our responses and comments are kept in black.

This technical report takes an interesting approach, harmonising multiple manuscript sources under a common framework and synthesising their findings into a unified indicator using various future projection results. It is a technical method of consolidating various types of data into a single metric and yields compelling results. In my opinion, the manuscript is ready for publication as a technical report.

While reviewing this manuscript, I came across several points that I found unclear. I have commented on these below.

minor comments

Lines 45–60, Section 2.1.2 and Figure 1: Please clarify the roles of what is represented as NDJFMA – xxx (e.g., DJF) and NDJFMA-decrease. My understanding is that equation (1) refers to NDJFMA-decrease, while Figure 1 shows NDJFMA – xxx. The decreases such as –25% mentioned in lines 58–60 presumably correspond to NDJFMA-decrease. It seems to me that NDJFMA – xxx and NDJFMA-decrease are conceptually different (the former being adjustments due to different averaging periods, and the latter being the actual future decrease ratio). However, in the current explanation, they appear to be mixed together. Could you please make their distinction more explicit?

We agree that this part may be misleading. We will clarify that literature values did not use a unique period to report seasonal decreases and depending on the period of interest those variable may vary significantly, making an intercomparison hard. We therefore synthesized these values to the NDJFMA-decrease and tried to highlight that a decrease in yearly snow depths is larger than decreases in winter snow depths. We will rewrite this paragraph accordingly and show a specific example using the decreases in Figure 2.

Lines 58–60: To which values do the reported decreases of 25% and 20% refer? They do not appear to be within the range shown in Figure 1. Could you please clarify what these percentages are based on?

We agree that this illustrative example is misleading as those values are not shown in Figure 1. We will change this example to a concrete example, using the decreases in Figure 2 and compute decreases for the different periods. We will additionally highlight those data points in Figure 1 for better understanding.

Lines 84–85: Could you include an illustration of $\Delta b$ and $\Delta c$ in Figure 2? It would help readers better understand the concept.

To illustrate $\Delta b$ and $\Delta c$ more clearly, we will add a line in peak snow depth to highlight $\Delta b$ and similarly for $\Delta c$. Additionally, we will add the following information in the legend for clarification: Future relative snow depth is computed by dividing the future snow depth by reference snow depth. We will change Figure 2 and the corresponding text accordingly.

Lines 141–143: I understand that, due to global warming, the snowmelt season begins earlier, as does the peak in snow depth. One point I found questionable is that the dependence of $\Delta b$ on elevation appears stronger than its dependence on temperature change compared to parameters such as a or $\Delta c$. The weak temperature dependence may be due to discontinuous changes; for example, when two peaks exist and the position of the dominant peak

shifts. However, the fact that Δb shows stronger elevation dependence than dependence on temperature change raises the question of whether this behaviour is a general characteristic or a result specific to the dataset used. If the latter, the explanatory power of the Δb equation would be reduced. It is important to clarify this point.

That is an interesting remark and we were also investigating this shift in more detail. We think it's important to keep in mind that with increasing elevation, the snow depth peaks later in the season as the accumulation period is longer. We further want to remind that we didn't use a specific dataset but in total 5 independent studies (see Table 1a in the appendix) for deriving this variable. That said, we argue that this variable Δb is important to preserve the local climatology rather than providing explanatory power related to temperature change. Technically, this simply means that the peak of the reduction curve is positioned relative to the peak in snow depth rather than fixing the curve to a specific date in the season. We will clarify the role of Δb in the manuscript.

---

## Author Comment (AC2)

**Author responses to J. Ignacio López-Moreno, reviewer 1 comments on:**

**"Technical note: Literature based approach to estimate future snow"**

by Richter et al. in *Hydrology and Earth System Sciences* (*HESS*)

We thank J. Ignacio López-Moreno for the time to assess our work and for the valuable feedback and suggestions. We respond to each point of the reviews below. The reviewer comments are highlighted in blue while our responses and comments are kept in black.

I enjoyed reading this note and believe it addresses, in a very smart way, an important issue in comparing previous snow projections: the use of different time horizons, models, emission scenarios, etc. Most of the implications of the assumptions and simplifications are well discussed. The manuscript is well written, and I did not identify any methodological flaws. Therefore, I recommend its publication.

Below, I provide a few minor suggestions and some ideas from my related research, which the authors may consider using to further strengthen the discussion:

-I wonder about the impact of the methodology used in previous studies to perturb observed series with climate projections (e.g., the Delta method on seasonal or monthly bases, quantile perturbation, or directly using simulated climate to drive snow models). Different methods may influence the asymmetry in the start and end of the snow season or other metrics that relate snow changes solely to temperature.

That's an interesting question, and we agree that the method used may have an impact on some snow metrics. However, we were unable to identify any clear influence of this aspect on the corresponding results. Probably, other differences between the studies and their uncertainties as well as the low number of studies prevented a clear conclusion in this regard.

- It is somewhat surprising to me that the changes in the start and end of the snow season appear symmetric. Is the projected temperature increase generally similar for winter and spring? Even if it is, I would expect some patterns related to elevation—for instance, an earlier snowmelt may eliminate periods of very high solar radiation, whereas a later snow onset may have less significant implications for incoming solar radiation and melt dynamics. This is particularly true at higher elevations but not at lower ones.

Unfortunately, we are not sure if we understand your question correctly since the projected snow depth changes for the start and end of the season are clearly asymmetric as shown in Fig. 2 (upper graph) and Figure 4. The reduction curve in Figure 2 (lower graph) is indeed symmetric but only around the value b. We didn't observe any consistent trend in their shape across the different studies. We will clarify this more explicitly in the revised manuscript in section 2.2 (lines 67-70).

-One of the strengths of this methodology is its ability to translate different scenarios into temperature changes. This can make it easier to communicate results for policy decisions, as many greenhouse gas emission targets are linked to temperature thresholds (e.g., 1.5 °C). This aspect could be highlighted more explicitly in the discussion, as it helps make the results more accessible to non-scientific audiences.

Thanks for this input on the additional strengths of our manuscript, we will add this extra benefit of our translation to temperature scenarios in the discussion.

- Related to the previous point, in recent years I have preferred, instead of simulating future snow conditions for different climate models and emission scenarios, to perform sensitivity analyses (e.g., adding 1–2 °C, or ±5–10 % changes in precipitation; see DOI: 10.1088/1748-9326/abb55f). Then, the changes in T and P can be framed with climate projections for specific regions. While this approach requires simplifying assumptions about the system, I find it makes the results much easier to compare. It may be interesting to contrast your approach with this type of sensitivity analysis.

Interesting point. We will add a few sentences in discussion to demonstrate the differences of the two approaches.

- Perhaps Figure 2 could more clearly illustrate how changes in "b" and "c" are derived.

To illustrate changes in "b" and "c" more clearly, we will add the following information in the legend: Future relative snow depth is computed by dividing the future snow depth by reference snow depth. Furthermore, we will add a line in peak snow depth to highlight Delta b and similar for Delta c. We will change Figure 2 accordingly and adapt the corresponding text.

Hoping my comments will result useful,

Best,

J. Ignacio López-Moreno

---

## Author Response (AR1)

**Author responses to J. Ignacio López-Moreno, reviewer 1 comments on:**

**"Technical note: Literature based approach to estimate future snow"**

by Richter et al. in *Hydrology and Earth System Sciences* (*HESS*)

We thank J. Ignacio López-Moreno for the time to assess our work and for the valuable feedback and suggestions. We respond to each point of the reviews below. The reviewer comments are highlighted in blue while our responses and comments are kept in black.

I enjoyed reading this note and believe it addresses, in a very smart way, an important issue in comparing previous snow projections: the use of different time horizons, models, emission scenarios, etc. Most of the implications of the assumptions and simplifications are well discussed. The manuscript is well written, and I did not identify any methodological flaws. Therefore, I recommend its publication.

Below, I provide a few minor suggestions and some ideas from my related research, which the authors may consider using to further strengthen the discussion:

-I wonder about the impact of the methodology used in previous studies to perturb observed series with climate projections (e.g., the Delta method on seasonal or monthly bases, quantile perturbation, or directly using simulated climate to drive snow models). Different methods may influence the asymmetry in the start and end of the snow season or other metrics that relate snow changes solely to temperature.

That's an interesting question, and we agree that the method used may have an impact on some snow metrics. However, we were unable to identify any clear influence of this aspect on the corresponding results. Probably, other differences between the studies and their uncertainties as well as the low number of studies prevented a clear conclusion in this regard. The corresponding uncertainties are mentioned in the 4$^{th}$ last paragraph in the discussion.

- It is somewhat surprising to me that the changes in the start and end of the snow season appear symmetric. Is the projected temperature increase generally similar for winter and spring? Even if it is, I would expect some patterns related to elevation—for instance, an earlier snowmelt may eliminate periods of very high solar radiation, whereas a later snow onset may have less significant implications for incoming solar radiation and melt dynamics. This is particularly true at higher elevations but not at lower ones.

Unfortunately, we are not sure if we understand your question correctly since the projected snow depth changes for the start and end of the season are clearly asymmetric as shown in Fig. 2 (upper graph) and Figure 4. The reduction curve in Figure 2 (lower graph) is indeed symmetric but only around the value b. We didn't observe any consistent trend in their shape across the different studies. We clarified this more explicitly in the revised manuscript in section 2.2.

-One of the strengths of this methodology is its ability to translate different scenarios into temperature changes. This can make it easier to communicate results for policy decisions, as many greenhouse gas emission targets are linked to temperature thresholds (e.g., 1.5 °C). This aspect could be highlighted more explicitly in the discussion, as it helps make the results more accessible to non-scientific audiences.

Thanks for this input on the additional strengths of our manuscript, we added a paragraph highlighting this extra benefit communication with non-scientific audience in the discussion.

- Related to the previous point, in recent years I have preferred, instead of simulating future snow conditions for different climate models and emission scenarios, to perform sensitivity analyses (e.g., adding 1–2 °C, or ±5–10 % changes in precipitation; see DOI: 10.1088/1748-9326/abb55f). Then, the changes in T and P can be framed with climate projections for specific regions. While this approach requires simplifying assumptions about

the system, I find it makes the results much easier to compare. It may be interesting to contrast your approach with this type of sensitivity analysis.

Interesting point. Our simple approach does only indirectly consider the impact of precipitations changes and does to assess the individual contribution of those drivers. According to climate models, the expected changes in winter precipitation are highly uncertain, and sensitivity studies show that temperature is clearly the dominant factor in the Alps. We agree that the described sensitivity approach has advantages, but it was never the purpose of our study to evaluate the individual contribution of the drivers. We added some sentences to explain this point in the discussion

- Perhaps Figure 2 could more clearly illustrate how changes in "b" and "c" are derived.

To illustrate changes in "b" and "c" more clearly, we marked Δb and Δc in the Figure 2 and adapted the legend: "The fitting parameter a, Δb and Δc are marked in orange, where a corresponds to the maximum future relative snow depth, Δb is the position of the peak of the reduction curve relative to the peak in reference snow depth, and Δc is the fraction of future season length to reference season length." Furthermore, we adapted the corresponding text in section 2.2..

Hoping my comments will result useful,

Best,

J. Ignacio López-Moreno

**Author responses to reviewer 2 comments on:**

**"Technical note: Literature based approach to estimate future snow"**

by Richter et al. in *Hydrology and Earth System Sciences* (*HESS*)

We thank the reviewer for the time to assess our work and for the valuable feedback and suggestions. We respond to each point of the reviews below. The reviewer comments are highlighted in blue while our responses and comments are kept in black.

This technical report takes an interesting approach, harmonising multiple manuscript sources under a common framework and synthesising their findings into a unified indicator using various future projection results. It is a technical method of consolidating various types of data into a single metric and yields compelling results. In my opinion, the manuscript is ready for publication as a technical report.

While reviewing this manuscript, I came across several points that I found unclear. I have commented on these below.

minor comments

Lines 45–60, Section 2.1.2 and Figure 1: Please clarify the roles of what is represented as NDJFMA – xxx (e.g., DJF) and NDJFMA-decrease. My understanding is that equation (1) refers to NDJFMA-decrease, while Figure 1 shows NDJFMA – xxx. The decreases such as –25% mentioned in lines 58–60 presumably correspond to NDJFMA-decrease. It seems to me that NDJFMA – xxx and NDJFMA-decrease are conceptually different (the former being adjustments due to different averaging periods, and the latter being the actual future decrease ratio). However, in the current explanation, they appear to be mixed together. Could you please make their distinction more explicit?

We agree that this part was misleading. We clarified that literature values did not use a unique period to report seasonal decreases and depending on the period of interest those variable may vary significantly, making an intercomparison hard. We therefore synthesized these values to the NDJFMA-decrease and tried to highlight that a decrease in yearly snow depths is larger than decreases in winter snow depths. We rewrote this paragraph in Section 2.1.2 accordingly and describe a specific example using the decreases in Figure 2 for better illustration.

Lines 58–60: To which values do the reported decreases of 25% and 20% refer? They do not appear to be within the range shown in Figure 1. Could you please clarify what these percentages are based on?

We agree that this illustrative example was misleading as those values are not shown in Figure 1. As written in the previous answer, we changed this example to a concrete example, using the decreases in Figure 2 and compute decreases for the different periods.

Lines 84–85: Could you include an illustration of Δb and Δc in Figure 2? It would help readers better understand the concept.

We added lined to highlight Δb and Δc in Figure 2 and adapted the legend: "The fitting parameter a, Δb and Δc are marked in orange, where a corresponds to the maximum future relative snow depth, Δb is the position of the peak of the reduction curve relative to the peak in reference snow depth, and Δc is the fraction of future season length to reference season length." Furthermore, we adapted the corresponding text in section 2.2.

Lines 141–143: I understand that, due to global warming, the snowmelt season begins earlier, as does the peak in snow depth. One point I found questionable is that the dependence of Δb on elevation appears stronger than its dependence on temperature change compared to parameters such as a or Δc. The weak temperature dependence may be due to discontinuous changes; for example, when two peaks exist and the position of the dominant peak shifts. However, the fact that Δb shows stronger elevation dependence than dependence on temperature change raises the question of whether this behaviour is a general characteristic or a result specific to the dataset used. If the latter, the explanatory power of the Δb equation would be reduced. It is important to clarify this point.

That is an interesting remark, and we were also investigating this shift in more detail. We think it's important to keep in mind that with increasing elevation, the snow depth peaks later in the season as the accumulation period is longer. We further want to remind that we didn't use a specific dataset but a relatively small number of in total 5 independent studies (see Table 1a in the appendix) for deriving this dependency. That said, we argue that this variable Δb is important to preserve the local climatology rather than providing explanatory power related to temperature change. Technically, this simply means that the peak of the reduction curve is positioned relative to the peak in snow depth rather than fixing the curve to a specific date in the season. We clarified the role of Δb in sections 2.2 and 3.1.